# Identification of Hit Compounds Using Artificial Intelligence for the Management of Allergic Diseases

**DOI:** 10.3390/ijms25042280

**Published:** 2024-02-14

**Authors:** Junhyoung Byun, Junhu Tai, Byoungjae Kim, Jaehyeong Kim, Semyung Jung, Juhyun Lee, Youn woo Song, Jaemin Shin, Tae Hoon Kim

**Affiliations:** 1Department of Otorhinolaryngology-Head & Neck Surgery, College of Medicine, Korea University, 02842 Seoul, Republic of Korea; 2Mucosal Immunology Institute, College of Medicine, Korea University, 02842 Seoul, Republic of Korea; 3Neuroscience Research Institute, College of Medicine, Korea University, 02842 Seoul, Republic of Korea

**Keywords:** artificial intelligence, allergic rhinitis, SOCS3, new drug development platform

## Abstract

This study aimed to identify and evaluate drug candidates targeting the kinase inhibitory region of suppressor of cytokine signaling (SOCS) 3 for the treatment of allergic rhinitis (AR). Utilizing an artificial intelligence (AI)-based new drug development platform, virtual screening was conducted to identify compounds inhibiting the SH2 domain binding of SOCS3. Luminescence assays assessed the ability of these compounds to restore JAK-2 activity diminished by SOCS3. Jurkat T and BEAS-2B cells were utilized to investigate changes in SOCS3 and STAT3 expression, along with STAT3 phosphorylation in response to the identified compounds. In an OVA-induced allergic rhinitis mouse model, we measured serum levels of total IgE and OVA-specific IgE, performed real-time PCR on nasal mucosa samples to quantify Th2 cytokines and IFN-γ expression, and conducted immunohistochemistry to analyze eosinophil levels. Screening identified 20 hit compounds with robust binding affinities. As the concentration of SOCS3 increased, a corresponding decrease in JAK2 activity was observed. Compounds **5** and **8** exhibited significant efficacy in restoring JAK2 activity without toxicity. Treatment with these compounds resulted in reduced SOCS3 expression and the reinstatement of STAT3 phosphorylation in Jurkat T and BEAS-2B cells. In the OVA-induced allergic rhinitis mouse model, compounds **5** and **8** effectively alleviated nasal symptoms and demonstrated lower levels of immune markers compared to the allergy group. This study underscores the promising nonclinical efficacy of compounds identified through the AI-based drug development platform. These findings introduce innovative strategies for the treatment of AR and highlight the potential therapeutic value of targeting SOCS3 in managing AR.

## 1. Introduction

The incidence of allergic rhinitis (AR) has steadily increased over the past few decades, with a notable prevalence in various regions reaching up to 40% [1]. Changes in environmental factors, such as the occurrence of haze, have contributed to a rapid rise in the prevalence of allergic diseases, including asthma, allergic dermatitis, and AR [2]. AR poses a global health challenge, significantly impacting both work and study. Within the European Union, the economic impact of AR is estimated to range from EUR 30 to 50 billion annually [3].

Allergic diseases manifest as type I hypersensitivity reactions and can be categorized into initial and later stages [4]. During the initial stage, T cells undergo differentiation into T helper 2 (Th2) cells, leading to the production of antigen-specific immunoglobulin E (IgE) in response to external antigen exposure. In the subsequent later stage, upon re-exposure to antigens, effector cells, including mast cells, release histamines, leukotrienes, and other substances, triggering allergic symptoms. This response is stimulated by cytokines, such as interleukin (IL)-4, IL-5, and IL-13, which are secreted by Th2 cells [5].

Current therapeutic approaches for allergies primarily target the numerous final mediators of anaphylaxis rather than addressing the underlying causes of the disease [6]. Immunotherapy, a notable and recent focus, involves incremental allergen injections to mitigate early-stage reactions and promote immune tolerance. However, this treatment entails significant costs and necessitates maintenance for over 3–5 years, potentially accompanied by severe side effects like anaphylactic shock [7]. A novel and prominent immunotherapeutic avenue explores the use of monoclonal antibodies. Nevertheless, this strategy is prohibitively expensive, and regular subcutaneous injections of monoclonal antibodies make it impractical for many patients [8].

Members of the suppressor of cytokine signaling (SOCS) family constitute a novel class of regulators for cytokine signal transmission through the Janus kinase (JAK)/signal transducer and activator of transcription (STAT) pathway. SOCS is involved in modulating Th1 and Th2 cytokine signal transmission, with SOCS3 directly inhibiting JAK kinase through its kinase inhibitory region [9,10]. Its implication in immune diseases, including allergic dermatitis and asthma, is notable, as overexpression of SOCS3 in T cells and B cells is correlated with Th2 reactions and blood IgE concentration [11,12]. Studies on SOCS3−/− transgenic mice have indicated a decrease in allergic reactions [13]. Beyond its kinase inhibitory region, SOCS3 features a SOCS box at the carboxyl (C)-terminal, bound by the ubiquitin ligase complex (E3). Degradation through the proteasome inhibits active JAK [14]. Targeting the core SOCS3 protein in cytokine regulation may be fundamental to treating allergic diseases, yet research in this field is limited, necessitating comprehensive investigation.

Moreover, an emerging strategy to streamline new drug development involves leveraging artificial intelligence (AI) to reduce time and costs in recent years [15]. AI applications span information exploration, candidate substance discovery, preclinical and clinical trials, and drug manufacturing, licensing, and usage [16]. Considering that only a fraction of clinical trial candidates advance to becoming new drugs, and the traditional development process takes 5–20 years with a cost exceeding USD 1 billion, AI shows that it has transformative potential. Utilizing AI in drug development could potentially cut development time to 3–4 years and reduce costs by over half [17,18,19,20].

Therefore, we aimed to utilize an AI-based drug development platform to synthesize new drug candidates targeting SOCS3. Our objective was to screen promising substances among the candidates and assess their effects at nonclinical levels, such as in vitro and in vivo. The findings of this study open new avenues for the development of therapeutic agents for allergic diseases.

## 2. Results

### 2.1. Hit Compounds Screened by AI

In this study, we identified hit compounds that target proteins by combining virtual screening methods based on computer-generated structures and biological evaluation. DB, converted into a 3D virtual ligand library, was used to conduct virtual screening of >8 million stereoisomers and to calculate mm/GBSA ΔG binding values. Afterward, the compounds that inhibit the SH2 domain binding activity of SOCS3 were screened using enzyme assays (Figure 1A). After molecular dynamics calculations of >200 compound complexes with higher docking scores, compounds with little fluctuation in the binding mode were ordered. Twenty hit compounds with excellent binding abilities were obtained after screening the AI-based drug inhibitory effects on the targeted sites (Figure 1B). Based on the effective ligand, a substance with better binding affinity was designed with drug characteristics. The remaining 264 compounds with a docking score of −7 and 414 compounds with a docking score of −6 were rejected because their inhibitory effects on the targets were relatively low compared to the selected 20 hit compounds.

### 2.2. Hit Compounds Inhibited SOCS3, Restoring JAK2 Activity without Harming Cell Viability

As shown in Figure 2A,B, compounds **5** and **8** showed no significant decrease in cell viability in BEAS-2B cells up to 100 µM. However, at 800 µM, the cell viability dropped to below 50%. In contrast, in Jurkat cells, compounds **5** and **8** exhibited cell viability rates of 84% and 87% at 100 uM, respectively. However, at 800 uM, the viability dropped to less than 10% (Figure 2C,D). With these results, it was observed that the selected compounds demonstrated relatively higher toxicity in immune cells compared to epithelial cells. Subsequent experiments took these data into consideration when determining drug concentrations.

We added SOCS3 into a tube containing PTK substrate and ATP and added JAK2 enzyme. The mixture was allowed to react for 45 min. After measuring luminescence and analyzing the results, we observed that JAK2 activity decreased with the increasing SOCS3 concentration (Figure 2E). When the concentration of SOCS3 was 0.1875 µM, the residual activity of JAK2 was 94%; when the concentration of SOCS3 was 1.5 µM, the residual activity of JAK2 was reduced to 44%; and when the concentration of SOCS3 reached 5.2 µM, the residual activity of JAK2 was further reduced to only 10%. The same method was used to determine the effectiveness of the hit compounds, according to the luminescence after injecting the effective substances, and the results were compared with those of the control group. The results showed that among the 20 screened hit compounds, the hit compounds **5**, **8**, **9**, and **10** restored JAK2 activity that was reduced by SOCS3 (Figure 2F). Moreover, the efficacies of hit compounds **5** and **8** were particularly significant. When 100 µM of hit compound **5** or **8** was added, the activity of JAK2, which was reduced by SOCS3 to approximately 25%, recovered to >50%. However, other hit compounds exhibited relatively poor efficacy and could only restore partial JAK2 activity.

### 2.3. The Hit Compounds Decreased SOCS3 Expression and Restored STAT3 Phosphorylation in Jurkat T and BEAS-2B Cells

After Jurkat T and BEAS-2B cell lines were stimulated with IL-6, the expression of SOCS3 and STAT3 and the phosphorylation of STAT3 at 5 min, 0.5, 1, and 2 h was confirmed using Western blot analysis. The efficacy of hit compounds **5** and **8** was also analyzed. The results showed that in the Jurkat T cell line, hit compound **8** could successfully inhibit SOCS3, especially from 0.5 h to 1 h, thus recovering the phosphorylation status of STAT3 (Figure 3A). With an increase in the hit compound concentration, SOCS3 inhibitory ability also increased. However, the SOCS3 inhibitory ability of the hit compound disappeared after 2 h. In the BEAS-2B cell line, hit compounds **5** and **8** could successfully inhibit SOCS3, especially at the time and concentration of 1 h and 100 µM, respectively, thus recovering the phosphorylation status of STAT3 (Figure 3B). At 5 min and 0.5 h, the efficacy of hit compound was not observed because SOCS3 had not yet been synthesized.

### 2.4. The Hit Compound Inhibited Systemic Allergic Responses in the Allergic Rhinitis Mouse Model

Before mice were sacrificed, a random observer recorded nasal rubbing and sneezing in each group for 15 min. The results showed that, although there was no statistical significance, nasal rubbing (Figure 4A) and sneezing (Figure 4B) frequencies of mice in the hit compound **5** and **8** treatment groups were reduced compared with those in the control group. The mouse total IgE and OVA-specific IgE were measured in mouse serum using ELISA, and the results showed that the total IgE in hit compounds **5** and **8** groups was reduced compared with the control group (Figure 4C). OVA-specific IgE levels in the hit compound **8** group were reduced compared to those in the control group (Figure 4D).

### 2.5. The Hit Compound Inhibited Local Allergic Responses in the Allergic Rhinitis Mouse Model

Gene expression in mouse nasal mucosa samples was measured using qRT-PCR, and it was found that IL-13 levels were significantly reduced in the treated groups. Although IL-5 and IL-4 levels were reduced in the treatment groups (Figure 5A–C), the difference was not significant. Histology analysis showed that, compared with the allergy group, eosinophil counts decreased in the nasal mucosa (Figure 5D) and lungs (Figure 5E) in the drug treatment groups.

## 3. Discussion

In this study, we effectively utilized a novel AI drug development platform to identify potential compounds capable of inhibiting SOCS3. Upon treating cells with the screened compounds, we observed a reduction in SOCS3 expression and the restoration of STAT3 phosphorylation. Furthermore, in vivo experiments demonstrated that these compounds decreased the frequency of nasal rubbing and sneezing in a mouse model of OVA-induced AR. Analysis of the drug treatment groups revealed lower levels of serum total IgE, OVA-specific IgE, and Th2 cytokines, including IL-4, IL-5, IL-13, IFN-γ, and eosinophils, compared to the allergy group. This suggests a potential therapeutic effect of the hit compounds on allergic responses. Importantly, our study demonstrated the successful identification of substances capable of inhibiting SOCS3 expression, referred to as hit compounds. Notably, the mechanism of action of these hit compounds differs from that of traditional drugs, which typically target endpoint substances like IgE and histamine in anaphylactic reactions [21]. Instead, our hit compounds act upstream in the signal transduction cascade, specifically inhibiting SOCS3, thereby preventing the initiation of allergic events at a fundamental level [12].

The SOCS protein family functions as cytoplasmic negative-feedback regulators, sharing similar structures and employing various mechanisms to negatively regulate cytokine signal transduction [22]. Among these mechanisms, SOCS family members can directly bind tyrosine kinases, leading to their inactivation. Alternatively, they can bind to cytokine receptors, occupying docking sites crucial for signal transduction mediation. Additionally, SOCS family members may act as components of E3 ligases, thereby regulating the ubiquitination and proteasomal degradation of target substrates [23].

Numerous studies have explored the relationship between SOCS3 and AR in clinical patients and animal experiments [24,25]. Previous experiments conducted by our team indicated that SOCS3 is upregulated in persistent AR, suggesting the potential significance of SOCS proteins as regulators in AR pathogenesis [26]. In comparison to a healthy control group, the expression of SOCS3 in the nasal mucosa and peripheral blood mononuclear cells of patients with persistent AR was notably increased. SOCS3 in healthy and allergic nasal mucosa typically localizes to the epithelium, submucosal gland, and endothelium. However, in persistent allergic nasal mucosa, SOCS3 exhibits greater staining intensity than in healthy nasal mucosa. Some studies have linked SOCS3 to mucus secretion in AR, demonstrating significantly increased SOCS3 protein expression in AR nasal mucosa.

In an AR mouse experiment, agomiR30a-5p, which inhibits SOCS3 expression, reduced mouse serum OVA-specific IgE concentration and the mRNA and protein levels of SOCS3 [27]. Another study utilized SOCS3 siRNA intranasal therapy in an asthma mouse model, resulting in decreased eosinophil counts, normalization of high reactivity to methacholine, improved mucus secretion, and reduced airway remodeling [28]. While this study focused on asthma, both AR and asthma fall under the category of united airway disease, making the findings relevant to the study of SOCS3 and AR [29]. In another study involving an asthma mouse model, matrine was found to inhibit SOCS3 expression, significantly reducing inflammatory cell infiltration, goblet cell differentiation, and mucus production in a dose-dependent manner. Matrine also decreased the expression of IL-4 and IL-13, aligning with our study’s results, where AI-screened hit compounds reduced the expression levels of IL-4 and IL-13 in our OVA-induced AR mouse model [30].

IgE plays a crucial role in mediating type I hypersensitivity, contributing to the pathogenesis of allergic diseases such as asthma and AR [31]. Despite its importance in immunobiology and allergic pathogenesis, the exact role of IgE remains unclear [32]. Our experimental findings in an OVA-induced AR mouse model demonstrated that the expression of total IgE or OVA-specific IgE was reduced with the administration of the identified hit compounds. This suggests promising potential for the treatment of AR.

Eosinophils, integral to Th2-driven inflammation, play significant roles in both stable and pathological conditions [33]. They are particularly involved in the pathogenesis of AR and asthma [34]. In chronic airway inflammation mediated by eosinophils, alarm cytokines IL-25, IL-33, and thymic stromal lymphopoietin (TSLP) from the epithelium induce local IL-5 production, a key molecule in eosinophil proliferation [35,36]. IL-5, along with various local mediators, triggers the release of toxic granule proteins through a secretion mechanism [37]. IL-4 and IL-13 contribute to the selective recruitment of eosinophils by inducing the expression of adhesion molecules in the endothelium [38]. Our experimental results revealed a decrease in the expression of IL-5, in addition to the previously mentioned reductions in IL-4 and IL-13. Histological analysis indicated a significant decrease in eosinophil numbers in the lung and nasal mucosa of mice treated with hit compounds, suggesting a substantial reduction in allergic reactions.

Th2 cytokines, including IL-4 and IL-13, play pivotal roles in AR [39,40]. An imbalance between Th1 and Th2 cytokines is associated with the inflammatory pathway in AR [41]. It has been proposed that a dysfunctional Th1 reaction forms the basis for more severe Th2 reactions in AR patients [42]. Th1 cells produce IL-2 and IFN-γ in response to allergic inflammation. Th1 cytokines can contribute to the destruction of the nasal epithelial barrier, leading to allergic inflammation [43]. The Th1 response, characterized by IFN-γ production, enhances immunity against intracellular pathogens and viral infections [44,45]. Our experimental results demonstrated that the identified hit compounds significantly reduced IFN-γ expression in the AR mouse model, holding great significance for the treatment of AR.

In this study, the compounds targeting SOCS3, identified through a novel AI drug development platform, exhibited remarkable efficacy at the preclinical stage. The screened hit compounds demonstrated the ability to alleviate symptoms associated with AR and reduce cytokine levels in an AR mouse model. The promising outcomes from this non-clinical evaluation suggest that further drug screening and subsequent clinical trials hold the potential to yield more effective drugs for AR treatment. We anticipate that the findings of this study will contribute to the development of novel treatment strategies for AR.

## 4. Materials and Methods

### 4.1. Drug Screening

From the list of drug-like screening compounds that can be ordered at any time from Enamine, DB was converted into a 3D virtual ligand library for the virtual screening of over 8 million stereoisomers. A grid was generated utilizing the ProteinPrep module (Maestro version 12.0.012) from Schrodinger(New York, NY, USA), a company that provides software leveraging artificial intelligence (AI) and computer simulation technologies to support drug discovery and development. Subsequently, virtual screening of the SOCS3 protein binding site, associated with the SH2 domain, was conducted using Schrodinger’s Glide program version 5.7. The mm/GBSA ΔG binding values were calculated, taking into account physico-chemical parameters such as pH, alkalinity, hydrogen bonding, and ionic bonding. Molecular dynamics, including solubility influenced by water molecules and potential fluidity upon dissolution, were computed, leading to the selection of compounds with stable binding modes. A substance with improved binding affinity was designed based on the effective ligand, incorporating drug characteristics. All these processes were carried out using Schrodinger’s program on computers equipped with high-performance GPUs like NVIDIA’s RTX3090 and CPUs like Intel’s I9.

### 4.2. Detection of JAK2 and SOCS3 Inhibition Capacity

When the concentration of SOCS3 increases, JAK2 inhibition is determined by adding SOCS3 into a tube containing protein tyrosine kinase (PTK, 10 mg/mL) substrate and ATP (500 µM). JAK2 enzyme (2.5 ng/µL) was added to the tube and the mixture was allowed to react for 45 min. A Spectramax^®^ i3x Multi-Mode Microplate Reader (Molecular Devices, San Jose, CA, USA) was used to measure the luminescence, and Softmax^®^ Pro software version 7.0.2 was used to analyze the results. The inhibitory ability of the selected hit compounds on the target is verified if the observed JAK2 activity is reduced after SOCS3 is restored by the hit compounds. Using the same experimental method, we measured luminescence after injecting effective substances according to the concentration and analyzed the results in comparison with the control group.

### 4.3. Cell Culture

To study the changes in SOCS3 expression, Jurkat human T-lymphocyte and BEAS-2B cells were used, which were incubated with 10% fetal bovine serum (FBS), penicillin/streptomycin, 50 µM β-mercaptoethanol (Sigma-Aldrich, St. Louis, MO, USA), and 2 mM glutamine in RPMI 1640 (WELGENE, Gyeongsan, Republic of Korea) at 37 °C. To verify performance changes involving the SOCS3 downstream signaling cascade, the Jurkat cell line was activated with CD3 (10 µg/mL, BioLegend, San Diego, CA, USA) and CD28 (2 µg/mL, BioLegend, San Diego, CA, USA), then stimulated with IL-6 (100 ng/mL, Peprotech, Cranbury, NJ, USA). Expression of SOCS3 and STAT3 and phosphorylation of STAT3 at 5 min, 0.5 h, 1 h, and 2 h was confirmed using Western blotting.

### 4.4. Analysis of Cell Viability

Cells were cultured in a 96-well plate at 1.0 × 10^4^ cells per well. To determine the toxicity of hit compounds **5** and **8**, cells were treated with serial 2-fold dilutions starting from 800 µM. The cells were incubated at 37 °C for 24 h. After refreshing with media containing CCK-8 solution (Ez-cytox, DLS, Seoul, Republic of Korea), cells were incubated for 2.5 h at 37 °C. The absorbance was measured at 450 nm using a microplate reader (SpectraMax Plus384, Molecular Devices, San Jose, CA, USA).

### 4.5. OVA-Induced Allergic Rhinitis Mouse Model

Female BALB/c mice (4 weeks old) were purchased from Orient Bio Inc. (Seongnam, Republic of Korea) and bred under specific pathogen-free conditions. Animal experiments were approved by the Institutional Animal Care and Use Committee of Korea University (KOREA-2020-0073). AR was induced using ovalbumin (OVA; Sigma-Aldrich, St. Louis, MO, USA). Mice were sensitized with 25 μg of OVA emulsified with 25 μL of adjuvant (ImjectTM Alum, Thermo Scientific, Waltham, MA, USA) in 125 μL of PBS by intraperitoneal injection on days 0, 7, and 14. The mice were divided into three groups (n = 7 each): the OVA treatment group, candidate substance 5 (with OVA) treatment group, and candidate substance 8 (with OVA) treatment group. Mice were intranasally administered 500 μg OVA resuspended in 50 μL PBS. After challenge, 50 μM of candidate substances 5 or 8 was administered intranasally on days 21–27. A randomized observer recorded the frequencies of nasal rubbing and sneezing in each group for 15 min on day 28. The mice were then euthanized and samples collected.

### 4.6. Enzyme-Linked Immunosorbent Assay (ELISA)

Total IgE and OVA-specific IgE levels in mouse serum were measured using commercial ELISA kits (total IgE; BD Bioscience, San Jose, CA, USA, OVA-specific IgE; BioLegend, San Diego, CA, USA) according to the manufacturer’s protocol.

### 4.7. Immunohistochemistry

The mouse head and lungs were immediately fixed after surgery in 4% paraformaldehyde for paraffin embedding. Paraffin-embedded turbinate or mouse head blocks were cut to 4 μm thickness on a microtome and deparaffinized. For histological analysis, all sections were stained using an H&E Staining Kit (YD Diagnostics, Yongin, Korea). The sections were examined using an Olympus BX51 microscope (Tokyo, Japan). Data were captured using an Olympus DP72 microscope digital camera with DP2-BSW software version 2.2 (https://www.olympus-lifescience.com/en/support/downloads/dp2-bsw_ver0202_step/, accessed on 4 January 2024).

### 4.8. RT-qPCR

Total RNA was extracted from the mouse lung and nasal mucosa, lysed with QIAzol (Qiagen, Valencia, CA, USA), and serially treated with chloroform, isopropanol, and ethanol to purify RNA. cDNA was synthesized from RNA using a cDNA synthesis master mix (GenDEPOT, Katy, TX, USA).

### 4.9. Real-Time PCR

Gene expression in mouse tissue samples was evaluated using quantitative real-time polymerase chain reaction (qPCR). The prepared cDNA was amplified and quantified using SYBR Green master mix (Qiagen, Valencia, CA, USA). PCR was performed using a real-time thermal cycler system (TP850) (Takara, Shiga, Japan) with 50 cycles of a two-step reaction that consisted of denaturation at 95 °C for 15 s followed by annealing-extension at 60 °C for 45 s. Data were analyzed using the ΔΔCt method. PCR primer sequences were as follows: IL-13, forward 5′-TCA AGT GGC ATA GAT GTG GAA GAA-3′ and reverse 5′-TGG CTC TGC AGG ATT TTC ATG-3′; interferon (IFN)-γ, forward 5′-CAG CAA CAG CAA GGC GAA AAA GG-3′ and reverse 5′-TTT CCG CTT CCT GAG GCT GGA T-3′; IL-4, forward 5′-ACA GGA GAA GGG ACG CCA T-3′ and reverse 5′-GAA GCC CTA CAG ACG AGC TCA-3′; IL-5, forward 5′-AGG CTT CCT GTC CCT ACT CAT-3′ and reverse 5′-TAC CCC CAC GGA CAG TTT G-3′; GAPDH, forward 5′-TTC AAC AGC AAC TCC CAC TC-3′ and reverse 5′-TCC TTG GAG GCC ATG TAG G-3′.

### 4.10. Western Blotting

SOCS3 expression in Jurkat T cells was detected using Western blotting. Protein extracts were separated using 10% sodium dodecyl sulfate-polyacrylamide gel electrophoresis and transferred to nitrocellulose membranes. Membranes were incubated with anti-AREG (Santa Cruz Biotechnology, Dallas, TX, USA), anti-TPST-1 (Abcam, Cambridge, UK), and anti-FLNB antibodies (Santa Cruz Biotechnology, Dallas, TX, USA), with anti-GAPDH antibodies (Santa Cruz Biotechnology, Dallas, TX, USA) as a reference. Protein visualization was performed using a ChemiDoc imaging system (Bio-Rad, Hercules, CA, USA).

### 4.11. Statistical Analysis

Statistical analysis was conducted using SPSS (v.20.0; IBM Corp., Armonk, NY, USA). Clinical characteristics or experimental data were calculated using Student’s t-test. *p*-values < 0.05 were considered significant.

## Figures and Tables

**Figure 1 ijms-25-02280-f001:**
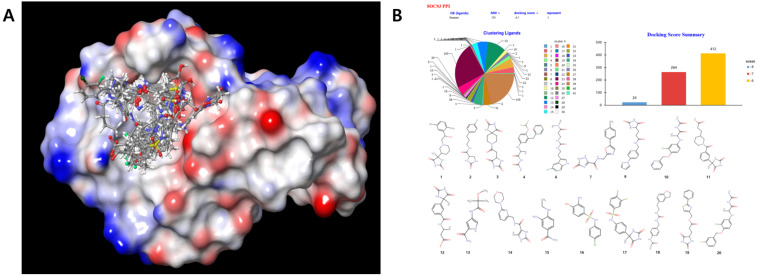
Virtual screening by AI using a virtual ligand database for the target combination site and 20 hit compounds with outstanding binding ability. (**A**) DB converted into a 3D virtual ligand library was used to conduct virtual screening of >8 million stereoisomers and to calculate mm/GBSA ΔG binding values, afterwards, the compounds that inhibit the SH2 domain binding of SOCS3 were screened. (**B**) After molecular dynamics calculations, compounds with little fluctuation in the binding mode were ordered. After screening their drug inhibitory effects on the targeted sites based on AI, 20 hit compounds with excellent binding ability were identified.

**Figure 2 ijms-25-02280-f002:**
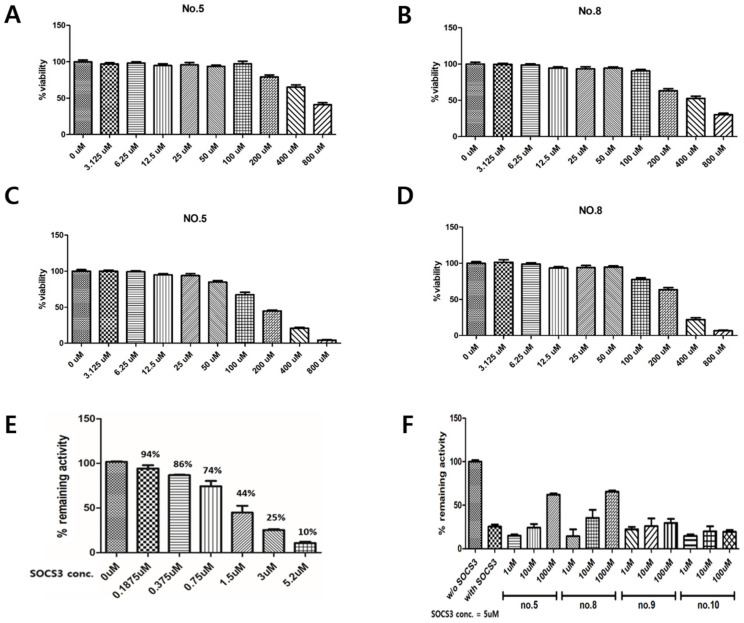
The hit compounds exhibited non-toxicity in both BEAS-2B and Jurkat cells, while restoring the suppressed JAK2 activity induced by SOCS3. (**A**) At concentrations up to 100 µM, compound **5** demonstrated no significant decrease in cell viability in BEAS-2B, but at 800 µM, the viability dropped below 50%. (**B**) At concentrations up to 100 µM, compound **8** demonstrated no significant decrease in cell viability in BEAS-2B, but at 800 µM, the viability dropped below 50%. (**C**) At concentrations up to 100 µM, compound **5** demonstrated no significant decrease in cell viability in Jurkat cell, but at 800 µM, the viability dropped below 10%. (**D**) At concentrations up to 100 µM, compound **8** demonstrated no significant decrease in cell viability in Jurkat cell, but at 800 µM, the viability dropped below 10%. (**E**) When the concentration of SOCS3 reached 5.2 µM, the residual activity of JAK2 was 10%. (**F**) When 100 µM of hit compounds **5** or **8** was added, the activity of JAK2, which was reduced from SOCS3 to approximately 25%, recovered to >50%.

**Figure 3 ijms-25-02280-f003:**
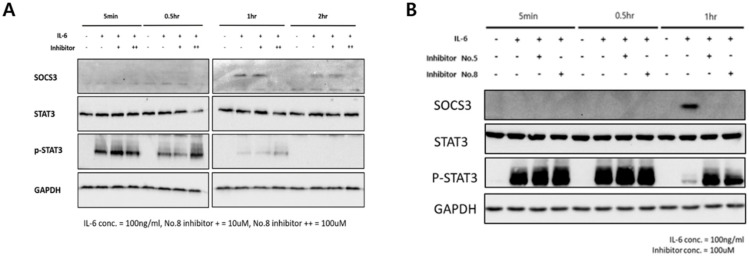
In Jurkat T and BEAS-2B cells, SOCS3 expression was reduced after treatment with hit compounds, and STAT3 phosphorylation was recovered. (**A**) In the Jurkat T cell line, hit compound 8 could inhibit SOCS3, from 0.5 h to 1 h, thus recovering phosphorylation of STAT3. (**B**) In the BEAS-2B cell line, hit compounds **5** and **8** could inhibit SOCS3 at 1 h, thus recovering phosphorylation of STAT3. Results presented in the graph are from three independent experiments.

**Figure 4 ijms-25-02280-f004:**
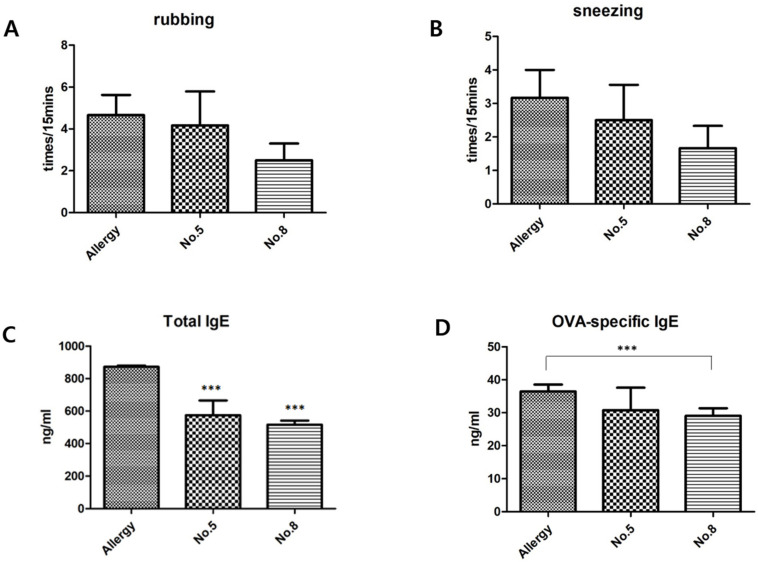
Effects of novel drug treatment on mouse symptom scores and immunoglobulin E levels in comparison to allergy group. (**A**) Nasal rubbing frequency of mice was reduced in hit compounds **5** and **8** treatment groups. (**B**) Nasal sneezing frequency of mice was reduced in hit compounds **5** and **8** treatment groups. (**C**) Expression of total IgE was reduced in hit compounds **5** and **8** treatment groups. (**D**) OVA-specific IgE expression was reduced in compound **8** group. ***: *p* < 0.001.

**Figure 5 ijms-25-02280-f005:**
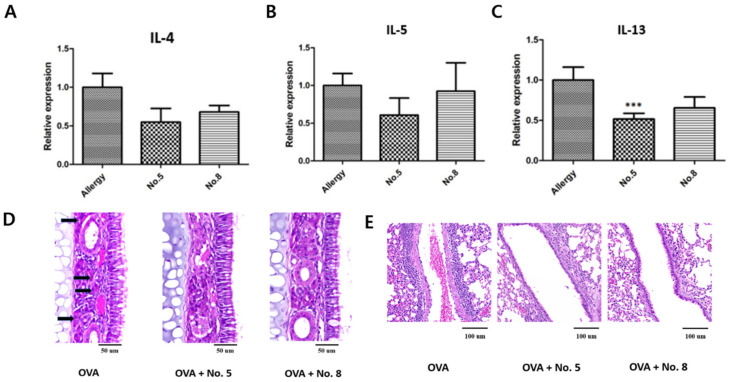
Reduction in eosinophils in mouse nasal mucosa and lungs in drug treatment groups compared to allergy group. (**A**) Expression of IL-13 was reduced in treatment groups. (**B**) Although the expression of IL-5 was also reduced in treatment groups, the difference was not statistically significant. (**C**) Although the expression of IL-4 was reduced in treatment groups, the difference was not statistically significant. (**D**) Eosinophils in mouse nasal mucosa decreased in treated groups. (**E**) Eosinophil levels in mouse lungs decreased in treated groups. ***: *p* < 0.001.

## Data Availability

The datasets supporting the conclusions of this article are included within the article.

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
