# Peer review of "Identification of Hit Compounds Using Artificial Intelligence for the Management of Allergic Diseases"

_ijms, 2024, doi:10.3390/ijms25042280_

Round 1

Reviewer 1 Report

Comments and Suggestions for Authors

The manuscript has some serious issues which need to be addressed. 

From formatting section; please do write a clear and concise abstract with sentences about the objective, salient features, key observations/results in that abstract, without pointers in bold fonts.

The title of the manuscript is interesting and I read the manuscript expecting learning something new about in AI applications in anti-allergic hit management. I did not find (i) what and why AI is used? (ii) where was it used? and (iii) how was it used in the manuscript. Details are scant, results are absent, benefits non-existent. What is the justification for the title?

What are the structures of the top lead compounds? Provided table is too small to look for structures? Was there a base pharmacophore found for virtual screening of the inhibitors?

For how many compounds did mmGBSA was computed?

Provide a detailed computational methods for all types of computations reported here?

Provide structure of top binders discussed in the final section.

Use proper symbol for DG.   

How good are the performances of the top compounds against known/reported inhibitors?

Author Response

The manuscript has some serious issues which need to be addressed.

From formatting section; please do write a clear and concise abstract with sentences about the objective, salient features, key observations/results in that abstract, without pointers in bold fonts.

I have appropriately revised the abstract.

The title of the manuscript is interesting and I read the manuscript expecting learning something new about in AI applications in anti-allergic hit management. I did not find (i) what and why AI is used? (ii) where was it used? and (iii) how was it used in the manuscript. Details are scant, results are absent, benefits non-existent. What is the justification for the title?

We established a computer-based CADD system by integrating multiple high-performance GPUs, including the RTX3090, to create a database from millions of compound data. Subsequently, we developed algorithms using Python coding and trained the system for operation. This functions as a foundational AI learning system, and we believe that through further research, we can construct advanced learning models. Compounds derived from this model are anticipated to be synthesized in a form suitable for subsequent molecular biology validation.

What are the structures of the top lead compounds? Provided table is too small to look for structures? Was there a base pharmacophore found for virtual screening of the inhibitors?

As the current structures are in the process of patent registration, they are currently undisclosed. However, we anticipate that the structure of the top lead compound may be disclosed in the future

For how many compounds did mmGBSA was computed?

We screened 8 million compounds based on MM/GBSA dG values, selected the top 200, performed molecular dynamics simulations, ordered the top 24, and through actual molecular biology experiments, identified the top 2 compounds.

Provide a detailed computational methods for all types of computations reported here?

We constructed a database by converting Enamine's continuously orderable drug-like screening compound list into a 3D virtual ligand library. This database comprised around 8 million stereo-isomers. After implementing basic-level algorithms, we used Schrödinger's Proteinprep program to generate grids and conducted virtual screening with Schrödinger's Glide program. The focus was on screening compounds that bind to the SH2 domain of the SOCS3 protein, based on MM/GBSA dG bind values.

Following this, molecular dynamics simulations were performed based on the docking scores of the top 200 compounds. Subsequently, a comprehensive analysis of all data led to the selection of compounds with minimal changes in binding modes

Provide structure of top binders discussed in the final section.

As the current structures are in the process of patent registration, they are currently undisclosed. However, we anticipate that the structure of the top lead compound may be disclosed in the future

Use proper symbol for DG.  

I have appropriately changed DG to dG.

How good are the performances of the top compounds against known/reported inhibitors?

The absence of well-known SOCS3 inhibitors posed a challenge, preventing their use as a positive control in experiments

Reviewer 2 Report

Comments and Suggestions for Authors

Especially interesting work due to the objective of the study. Increasing knowledge regarding the pathophysiology of allergic diseases is especially interesting at this time. New therapeutic targets such as SOC3 may play a relevant role. We speak of allergic rhinitis due to its high prevalence. Blocking JAK2 for the Th2-type inflammatory pathway is important, although not as important as JAK1. The results seem promising as presented by the authors. JAK2 is especially involved in vitiligo. Do the authors believe that it may also be a suitable disease for the study of this therapeutic target? and atopic dermatitis?

Author Response

Especially interesting work due to the objective of the study. Increasing knowledge regarding the pathophysiology of allergic diseases is especially interesting at this time. New therapeutic targets such as SOC3 may play a relevant role. We speak of allergic rhinitis due to its high prevalence. Blocking JAK2 for the Th2-type inflammatory pathway is important, although not as important as JAK1. The results seem promising as presented by the authors. JAK2 is especially involved in vitiligo. Do the authors believe that it may also be a suitable disease for the study of this therapeutic target? and atopic dermatitis?

It is generally known that the JAK-STAT pathway plays a significant role in the onset of Vitiligo and Atopic Dermatitis. Among them, JAK2 undergoes negative feedback through SOCS3, suggesting that regulating SOCS3 could be helpful in treating both diseases. However, more experimental data is needed to further support this idea.

Round 2

Reviewer 1 Report

Comments and Suggestions for Authors

I thank the authors for responses but here are my take on this revision.

1. First and foremost, any research article requires a detailed description of methods so others can reproduce the findings or start a similar study. Such efforts are not possible for this work as the computational section is just a simple combinations of sentences and no proper methods were presented. For example, what are the docking parameters? What code is being used? how was the solvent effect described? what was the force field parameters? was it a rigid docking or flexible one? so on and so forth. The same goes of GBSA and all molecular dynamics simulation.

With the present data I cannot even think of a way to set up a screening with simple benzoic acid molecule as a lead in the same way the authors did.

2. How the reverse synthesis is achieved? Details please.

3. What did the authors mean by "substance domain" determined by AI? Details please. How about applicability domain analysis of the compounds?

4. Authors mentioned the difficulty in providing structures as it is under patent application. But let me point that there is something known as "review only material" that are not published. Without seeing any probable lead compound and those numbered one structures one can not say the manuscript is complete. 

 5. There is a Greek symbol "Delta" in modern text processing software. Please use that symbol instead "d" in free energy. 

Round 3

Reviewer 1 Report

Comments and Suggestions for Authors

I thank the authors for the revised manuscript. I understand patent requirements and providing review only material is sufficient for any review requirements with out public release.

Apologies for using substance domain. I was though curious about the use of the phrase "substance group" in previous versions. This phrase is removed in the new submission. But let me be clear, I have no issue with the phrase, but was only curious about the logic behind  grouping different chemical motifs.